# Integrated Cognitive Rehabilitation Home-Based Protocol to Improve Cognitive Functions in Multiple Sclerosis Patients: A Randomized Controlled Study

**DOI:** 10.3390/jcm11123560

**Published:** 2022-06-20

**Authors:** Minoo Sharbafshaaer, Francesca Trojsi, Simona Bonavita, Amirreza Azimi

**Affiliations:** 1MS Research Center, Neuroscience Institute, Tehran University of Medical Science, Tehran 1136746911, Iran; minoo.sharbafshaaer@unicampania.it (M.S.); a-azimi@sina.tums.ac.ir (A.A.); 2MRI Research Center SUN-FISM, Department of Advanced Medical and Surgical Sciences, Università degli Studi della Campania “Luigi Vanvitelli”, 80138 Naples, Italy; francesca.trojsi@unicampania.it

**Keywords:** multiple sclerosis, integrated cognitive rehabilitation, cognitive functions

## Abstract

Cognitive impairment (CI) occurs in about 40–65% of people with multiple sclerosis (MS) during the disease course. Cognitive rehabilitation has produced non-univocal results in MS patients. Objective: The present study aimed to evaluate whether an Integrated Cognitive Rehabilitation Program (ICRP) in MS patients might significantly improve CI. Methods: Forty patients with three phenotypes of MS were randomly assigned into two groups: the experimental group (EG, n = 20), which participated in the ICRP for 10 weeks of training; and the control group (CG, n = 20). All participants’ cognitive functions were assessed at three timepoints (baseline, post-treatment, and 3-month follow-up) with the California Verbal Learning (CVLT), Brief Visuospatial Memory (BVMTR), Numerical Stroop, and Wisconsin tests. Results: When compared to CG patients, EG patients showed significant improvements in several measures of cognitive performance after ICRP, including verbal learning, visuospatial memory, attention, and executive functions. Conclusions: Home-based ICRP can improve cognitive functions and prevent the deterioration of patients’ cognitive deficits. As an integrated cognitive rehabilitation program aimed at potentiation of restorative and compensatory mechanisms, this approach might suggest an effective role in preserving neuronal flexibility as well as limiting the progression of cognitive dysfunction in MS.

## 1. Introduction

Multiple sclerosis (MS) is a progressive inflammatory and immune-mediated neurological disease that causes central nervous system demyelinating disease [1,2]. Typically, the disease presents in young adults (mean age of onset, 20–30 years) [3]. Moreover, MS is associated with progressive brain atrophy and white and gray matter damage [4], which in turn correlates with clinical disabilities [5]. Cognitive impairment (CI) affects 40–65% of patients during the course of the disease [6,7]. CI can appear early in the course of MS, be unrelated to physical disability, and worsen over time [7]. Patterns of cognitive dysfunction are heterogeneous in MS, and this can be noticed in the early stages of the disease as well. Different phenotypes of MS are comorbid with cognitive impairment. More frequent and severe deficits are reported in the secondary-progressive phenotype (SPMS) than in the relapsing-remitting phenotype (RRMS) [8]. Specifically, CI could be present in all MS subtypes from the clinical onset, and its frequency is higher in the progressive forms [9]. Dackovic et al. [10] showed that among 168 MS patients, CI was more severe and frequent in those with SPMS, followed by those with primary-progressive MS (PPMS), and then by clinically isolated syndrome (CIS) and RRMS patients [10]. The cognitive domains recognized as mainly impaired in MS are memory, in its different dimensions [11], attention [12], and executive functions [13], allowing us to identify different phenotypes of cognitive impairments that represent a more meaningful concept of the cognitive status of people with MS [14]. Therefore, the performance of various daily activities is commonly compromised by cognitive deficits [15].

On the other hand, CI is a relatively less explored but crucial aspect of cerebral dysfunction in MS patients. It has been related to focal T2 hyperintense lesions, diffuse white matter (WM) damage, and cortical and deep gray matter (GM) atrophy [16]. Additionally, a more recent positron emission tomography (PET)-magnetic resonance imaging (MRI) study showed that lower myelin content with structural damage was associated with cognitive deficits [17,18]. It has been hypothesized that CI becomes evident when the dynamic balance between brain damage and brain reorganization collapses, so functional brain reorganization is not effective and clinical impairments may appear [19]. Therefore, revealing the factors that could facilitate care for the subpopulation of patients in whom CI occurs would be crucial. In this regard, considering that effective pharmaceutical treatments for CI are still lacking, cognitive rehabilitation and neuromodulation techniques using non-invasive approaches could help improve cognitive performance in people with MS [20].

Recently, several studies have been aimed at evaluating the efficacy of cognitive rehabilitation in MS patients [6,20,21]. Most interventions have included actions to improve memory, attention, and executive function [6,21,22]. Notably, Bonavita et al. [22] revealed that, after a short-term computer-based cognitive rehabilitation program, patients with relapsing-remittent (RR) MS and CI showed significant improvement in executive and memory performance together with a significant increase in functional connectivity in some posterior areas of the default mode network, as proven by resting-state functional MRI monitoring of brain functional connectivity before and after this rehabilitative intervention. These relevant findings suggest that cognitive rehabilitation may induce adaptive cortical reorganization favoring better cognitive performance in RRMS, thus strengthening the value of cognitive exercise for limiting CI [22]. However, despite the range of cognitive rehabilitative treatments available for MS, there is a lack of high-quality evidence for many approaches [21]. Remarkably, multidisciplinary and integrated rehabilitation programs that lead to longer-term gains at the levels of activity and participation may highly impact cognitive symptoms through more efficient recruitment of brain regions [23], thereby positively influencing MS patients’ daily lives [24]. Moreover, the effectiveness of integrated cognitive rehabilitation programs should be more efficient for MS patients through homework intervention, making the content of the goals of these interventions and homework tasks more ecological [25]. The cognitive improvements observed after the implementation of homework interventions indicate that this personalized cognitive training could be a practical and valuable tool to improve cognitive skills [26,27] and additionally enhance neuronal plasticity in MS patients [27], which is associated with modifications of functional and/or structural plasticity within specific brain networks/regions involving cognitive functions [28,29]. Furthermore, findings from several investigations revealed enhanced recruitment of brain networks serving trained functions in response to training/cognitive rehabilitation programs to compensate for damaged networks in patients with RRMS [22,27] or progressive MS [28,30].

To shed more light on the potential benefits of the Integrated Cognitive Rehabilitation Program (ICRP) on the cognitive performance of MS patients, we performed a multi-center, longitudinal study of cognitive function monitoring in MS patients undergoing ICRP, which included paper-pencil homework, cognitive game training, and physical exercise for three months. We hypothesized that, compared to patients in usual care, the MS patients who participate in the ICRP would show significantly greater improvements in verbal learning memory, visuospatial memory, attention, and executive function across time.

## 2. Materials and Methods

The statistical population in this study was people with multiple sclerosis (relapsing-remitting, primary-progressive, and secondary-progressive) [31] who were members of the MS Associations of Alborz, Zahedan, and Zabol, Iran. In addition, patients had at least two years of membership in this MS Association. In order to include only definite diagnoses, our population was limited to patients who had been diagnosed, started disease-modifying therapies (DMTs), and were followed up by various MS centers in different regions of Iran. As for sample size, by estimating from previous studies, such as the research by Bonavita et al. [22], Rilo et al. [32], and Filippi et al. [29], and considering the 95% confidence interval and 85% test power, we finally enrolled 40 consecutive MS patients in three cities who were randomly assigned by the random sampling method available into two groups, experimental (n = 20) and control (n = 20).Participants were enrolled in this research from 22 June 2020,aged between 18 years and 65 years, with a literacy level of no less than high school education (i.e., >12 years),and two years post-MSMS diagnosis according to the revised McDonald’s criteria including RRMS, PPMS, and SPMS phenotypes [31]; stable DMT; and Expanded Disability Status Scale (EDSS) score < 5.5, according to the neurological assessment [33,34]. Exclusion criteria included a history of mental retardation, major depression, or other neurological and psychiatric disorders; relapses and steroid treatment in the previous 6 months before enrollment and during the trial, and participation in similar research in the last 3 months as an intervention on cognitive functions. The effects of the intervention on outcomes were assessed over 1 month, with measurements at baseline, immediately after the 45 sessions (every 20 min or so for 10 weeks) of integrated cognitive rehabilitation intervention, and at 3 months post-intervention. The statistical significance level was set at *p <* 0.05. Please see Figure 1 for the flowchart of the study design.

### 2.1. Neuropsychological Assessments

#### 2.1.1. California Verbal Learning Test (CVLT) and Brief Visuospatial Memory Test (BVMTR)

The consensus was achieved on optimal measures for learning and memory in MS patients, time permitting: the initial learning trials of the second edition of the California Verbal Learning Test (CVLT) and the revised Brief Visuospatial Memory Test (BVMTR). These two scales comprised the Brief International Cognitive Assessment for MS (BICAMS). The auditory/verbal learning test in the CVLT begins with the examiner reading a list of 16 words with five learning trials, and patients listen to the list and report as many of the items as possible after a recall is recorded. Visual/spatial memory is assessed by BVMTR: according to the test, six abstract designs are presented for 10 s, and the display is removed from view. The patients render the stimuli via pencil-on-paper manual responses, and the scores range from 0 to 12 [35].

#### 2.1.2. Numerical Stroop

In a numerical Stroop experiment, participants perform a physical or a numerical size judgment task in separate blocks. In the numerical task, participants respond to the values and ignore the physical sizes. In neutral pairs, the two digits vary in one dimension only (e.g., the pair 5 3 for the numerical task and large 3 small 3 for the physical task). Neutral pairs enable measuring facilitation (i.e., the difference in reaction time between neutral and congruent pairs) and interference (i.e., the difference in reaction time between incongruent and neutral pairs) [36].

#### 2.1.3. Wisconsin Card Sorting Test (WCST)

Berg and Grant developed the Wisconsin Card Sorting Test (WCST) in 1948. It is considered a measure of executive function. WCST consists of four stimulus cards, placed in front of the subject: they depict a red triangle, two green stars, three yellow crosses, and four blue circles, respectively. The subject receives two sets of 64 response cards, which can be categorized according to color, shape, and number. The subject is told to match each of the response cards to one of the four stimulus cards and is given feedback on each trial on whether he/she is right or wrong [37].

### 2.2. Integrated Cognitive Rehabilitation Program (ICRP)

#### 2.2.1. General Considerations

The ICRP intervention aims at helping people with MS to acquire the highest level of cognitive functioning and functional independence. The intervention includes group sessions (2 h per week for 10 weeks) focused on building efficacy for use of cognitive strategies and home-based, paper-pencil, computer-game tasks, and physical function interventions (45 min three times per week). The Integrative Cognitive Rehabilitation Program offers a comprehensive, integrative, and multidisciplinary approach to help individuals develop compensatory strategies and maximize individual strengths; ICRP is challenging to improve patient outcomes [38].

The ICRP program was implemented in patients who conformed to the experimental group. The cognitive rehabilitation program is theoretically based on an individualized approach that is based on the individual’s strengths and works to compensate for deficit areas to increase the person’s ability to fully participate in daily life activities [39]. The ICRP follows a theoretical model that represents the flow and organization of cognitive rehabilitation provided in the ICRP program (see Figure 2).

#### 2.2.2. Paper-Pencil

Paper-and-pencil tasks are still the most widely used methods for cognitive rehabilitation because of their accessibility, ease of use, clinical validity, and reduced cost. In recent years, computer-based versions of these traditional tasks have also begun to become clinically accepted. Through computational modeling, the authors operationalized 11 paper-and-pencil tasks and developed an information and communication technologies (ICT)-based tool. The NeuroRehabLab Task Generator is used to tailor each of the 11 paper-pencil tasks (15 min of practice) in the domains of memory, attention, and executive functions to each patient [40].

#### 2.2.3. Computer-Game

A computer game with a mobile version is a novel tool for encouraging exercise, improving cognitive functions, and encouraging training. We included N-back, Neuro-active, Making words, Lumosity, and Memory-match. This protocol addresses the most common deficits experienced by people with MS (attention, memory, and executive functions). Each participant was asked to complete three sessions (45–60 min of training) a day, three times a week, (approximately 45 games), and to keep a written log of practice time and record the results of each game that they played. These participants also received weekly “check-in” calls from research staff during the 10 weeks of the intervention period [41], although a cognitive therapist was available to patients every day of the intervention’s weeks to receive their function reports.

#### 2.2.4. Physical Exercise Intervention

The physical exercise intervention would improve physical function by sending videos of practices that promoted home-based endurance and strength exercises. The exercise intervention was delivered remotely. Physical activity was objectively evaluated at study inception and progressively increased according to the patient’s abilities. Individual exercise prescriptions were individually tailored (“dosed”) to correspond to functional status levels and primarily targeted lower extremity function and endurance using exercises that could be easily performed at home (e.g., hand and finger practices, walking, and leg exercise). The exercise intervention included eight short videos (20–35 min for exercise). Each call followed a structured protocol to assess previously prescribed exercises, explore and address motivation and encourage continued exercise [32,42].

### 2.3. Statistical Analysis

The data were processed using SPSS/Windows, version 25. Descriptive statistics were used to explore the demographic and clinical characteristics of the participants. A repeated-measures ANOVA (2 (GE and GC) × 3: baseline assessment (T1), post-intervention (T2), and long-term evaluation (T3)) was performed to check the effect of the program in EG patients in comparison to CG patients. Repeated measures on one factor were conducted. Mauchly’s test of sphericity was used to assess whether the assumption of sphericity was met. The Greenhouse–Geisser effect correction was employed when sphericity was violated [43]. The effect sizes were calculated using omega-squared (ω2). The reference values for omega-squared were 0.01, 0.06, and 0.14 (small, medium, and large effect size, respectively). An additional independent-samples *t*-test was conducted to compare the differences in the scores recorded at the baseline and time 3 (*change score*). Effect sizes were calculated using Cohen’s *d*. Cohen classifies 0.2 as a small effect, 0.5 as a medium effect, and 0.8 as a large effect [44]. Values of *p* < 0.05 were considered significant.

## 3. Results

### 3.1. Characteristics of the Sample

#### 3.1.1. Comparison of Demographic and Clinical Characteristics at Baseline (Pretreatment)

In general, a higher proportion of females compared to males took part in the study. The percentage of females was higher in both groups (70% for the rehabilitation and 80% for the control group), something that was expected due to the higher female to male ratio in the MS population in general. However, the proportion/ratio of females between the two groups was not significantly different [*U* (39) = 149.00, *p* = 0.988]. The main age level in EG was the 31–43 _age group_ (45%) and the 18–30 _age group_ (50%) in the CG. However, there was not any significant difference between the two groups [*x*^2^ (1) = 0.512, *p* = 0.774]. The disease duration of 2–8 years was the same in both groups [*x*^2^ (1) = 4.489, *p* = 0.106]. No history of MS disease in the family [*U* (39) = 208.00, *p* = 0.072], the rate of phenotypes of MS [*x*^2^ (1) = 4.305, *p* = 0.116], and use of disease-modifying therapies (DMTs) [*U* (39) = 190.00, *p* = 0.892] were similar in both groups. Then, we investigated the normality distribution of our data with the Shapiro–Wilk normality test. The null hypothesis could not be rejected; therefore, we used the parametric independent samples t-test to test group differences in this variable. (See Table 1 for a detailed description of baseline demographic and clinical characteristics.) From the above analysis, we concluded that our two groups were well-matched on baseline demographic variables.

#### 3.1.2. Comparison of Neuropsychological Test Performance for the ICRP MS-Experimental Group between Baseline, Posttreatment, and at 3-Month Follow-Up

We found improvements in auditory memory/verbal learning (CVLT), visual/spatial memory (BVMTR), attention, and executive functions across three-time assessments (baseline, posttreatment, and 3-month follow-up) (Table 2). In addition, the time was positively related to most of the assessed performances from baseline to posttreatment. Mixed between-subject analysis of variance showed that the patients who received functional cognitive training had improved cognitive performance between baseline, post-test, and 3-month follow-up, and then, there was a substantial main effect of time on the CVLT [Wilks’ Lambda = 0.363, *F* (1, 38) = 38.816, *p* > 0.001, partial eta squared = 0.637, observed power = 1]. Additionally, significant interaction was revealed between program type and time [Wilks’ Lambda = 0.494, *F* (1, 38) = 38.816, *p* > 0.001, partial eta squared = 0.637, observed power = 1]. There was a significant effect of time on the BVMTR [Wilks’ Lambda = 0.089, *F* (1, 38) = 389.120, *p* > 0.001, partial eta squared = 0.911, observed power = 1]. Moreover, we observed a significant interaction between program type and time [Wilks’ Lambda = 0.281, *F* (1, 38) = 97.280, *p* > 0.001, partial eta squared = 0.719, observed power = 1]. There was a significant effect of time on attention [Wilks’ Lambda = 0.276, *F* (1, 38) = 48.623, *p* > 0.001, partial eta squared = 0.724, observed power = 1]. In addition, we revealed significant interaction between program type and time [Wilks’ Lambda = 0.339, *F* (1, 38) = 36.124, *p* > 0.001, partial eta squared = 0.661, observed power = 1]. There was also an effect of time on executive function [Wilks’ Lambda = 0.328, *F* (1, 38) = 37.913, *p* > 0.001, partial eta squared = 0.724, observed power = 1]. Finally, significant interaction was found between program type and time [Wilks’ Lambda = 0.339, *F* (1, 38) = 36.124, *p* = 0.001, partial eta squared = 0.661, observed power = 0.95] (for more details, see Table 3).

#### 3.1.3. Comparison of Neuropsychological Test Performance for the MS Standard Care Control Group between Baseline and Posttreatment

Our results revealed that in all measures, there were significant changes between pre- and post-treatment assessments. An exception was the performance on the mean auditory memory/verbal learning rate, which increased from *M*_baseline_ = 38.55 to *M*_posttreatment_ = 44.50 [z = 80, *p* = 0.000] (Figure 3), the mean visual/spatial memory rate, which improved from *M*_baseline_ = 2.80 to *M*_posttreatment_ = 4 [z = 72, *p* = 0.000] (Figure 4), attention, which increased from *M*_baseline_ = 0.427 to *M*_posttreatment_ = 0.626 s [z = 71.703, *p* = 0.000] (Figure 5), and executive function, which increased from *M*_baseline_ = 0.811 to *M*_posttreatment_ = 0.847 s [z = 72.521, *p* = 0.000] (Figure 6). These findings, although marginally different in some cases, produced statistically significant changes over time, with a positive direction. These results imply that this group showed improvements over time (see Table 4).

## 4. Discussion

In the present investigation, we conducted a 3-month ICRP controlled trial in order to restore the main cognitive domains impaired in MS patients. Our findings allowed us to differentiate between the two groups that took part in the study (EG, rehabilitation; and CG, control).

Following the home-based ICRP, results showed significant improvement in auditory/verbal learning (CVLT), visual/spatial memory (BVMTR), attention, and executive functions tests with patient reports. On follow-up assessment, findings revealed a significant increase in the “four dimensions” of cognitive performance.

The auditory/verbal learning, visual/spatial memory, attention, and executive functions increased after treatment without declining to pretreatment levels in the following 3 months after rehabilitation. When cognitive domain performance between the ICRP rehabilitation group and control group was compared over time, we noted that the rehabilitation group outperformed the control group on all derived domains from pre-to post-treatment assessments.

It is plausible to consider that MS patients performing home-based ICRP learned exercises and transferred the cognitive strategies to different aspects of daily life, and this may have induced more independence in patients’ lives. For this purpose, since integrated cognitive rehabilitation treatment has a significant role in improving the cognitive functions of patients, in addition to pharmacological treatment, regular and relevant cognitive rehabilitation programs should be considered for MS patients. Thus, home-based integrated cognitive rehabilitation may lead to flexibility, clinical efficiency, ecological benefits, sufficient/saving time, and create an interesting clinical option in the MS population that aims at improving cognition not through a specific neurological tool but by enhancing neurocognitive skills in patients’ daily lives over time. At baseline, training time was dedicated to cognitive functions, suggesting that the observed improvements in attention, visual/spatial memory, auditory/verbal learning, and executive function could be explained by the program’s assigning a relatively integrated cognitive training component. Interestingly, Shatil et al. [26] agreed about the effect of training time on memory and attention in MS patients, whereas home-based integrated cognitive rehabilitation 2–7 times a week could improve executive function and improve cognitive outcomes in people with MS [45].

Regarding the present research approach, other studies revealed similar findings, including Barbarulo et al. [38], who showed that rehabilitation strategies based on paper-pencil tasks can be personalized and adapted to MS patients’ daily lives; [46] they also suggested that software programs may be a valuable option for integrated cognitive rehabilitation of people with MS disease with improved auditory/verbal learning, visual/spatial memory, attention, and executive functions. Moreover, Jackson et al. [47] found that physical and functional training was feasible and possibly effective in memory, attention, and executive functions. In addition, Gómez-Gastiasoro et al. [48] emphasized the necessity and importance of designing and implementing integrated cognitive rehabilitation programs that enhance attention, executive functioning, and long-term visual and verbal memory in MS patients. Rilo et al. [32] reported results on patients who received cognitive rehabilitation for 3 months focused on training, learning, and implementing memory and executive function, showing moderate improvements in working memory, verbal memory, and executive function. Campbell et al. [49] supported the hypothesis that integrated home-based cognitive rehabilitation is a practical and effective approach to improving auditory/verbal learning and visual/spatial memory in people with MS and may reflect fundamental changes in brain activation. In particular, Ghahfarrokhi et al. [45] found that home-based rehabilitation, 2–7 times per week, is beneficial, feasible, and safe for people with MS. Finally, Centonze et al. [42] showed that physical exercise rehabilitation is more effective in improving patients’ cognitive functions and even their performance in cognitive tasks of training protocols.

Considering that ICRP for MS patients is a regular and functional set of therapeutic activities, cognitive improvement is achieved by reinforcing previously learned behavioral patterns as well as by creating new and compensatory patterns. This is probably due to the flexibility of the brain, triggering changes in brain organization that are possible under the influence of short-and long-term behavior modification. In this regard, some evidence suggests that beneficial effects are mediated by both immune modulation and activity-dependent plasticity in the brain [50]. Other researchers found a correlation between improved function and magnetic resonance imaging (MRI)-detected brain changes, thus supporting the hypothesis that training-induced brain plasticity is specifically linked to rehabilitation training [51]. These changes can be structural, with the reconstruction of the physical or physiological structures of the brain, or functional, with the dynamic regulation of brain connectivity. These structural and functional brain changes may be underpinned by molecular/cellular mechanisms that may explain improvements or brain network rearrangements [52], such as synapse formation that modulates impact transfer resistance. These changes can include adaptation to new conditions and different types of learning and compensatory changes in response to the impairment of cognitive function in MS patients [53].

To note, with the advent of modern technological advances in medical care, approaches to rehabilitation, treatment, and cognitive education are moving online and can be adapted and personalized. The main advantage of this method was that it allowed access to the intervention from home. Participants in ICRP from home may have the possibility of rapid presence in the intervention, a strong, low-cost program, and training in the real world [25].

Our study has some limitations. The major limitations are the small sample size; the heterogeneity of the studied population including three disease phenotypes (RRMS, PPMS, and SPMS); the inclusion of patients undergoing only DMTs, which have been shown to influence cognitive performances [54]; the short duration of rehabilitation training; and the short follow-up time without monitoring brain structural and functional changes by advanced neuroimaging techniques. Furthermore, other limitations are the lack of stratification of cognitive disorders to perform tailored interventions and the main use of “check-in” calls for patient monitoring. Future studies should be able to explore the guarantee and value of integrative cognitive rehabilitation by using specific structural and functional examinations (such as EEG and fMRI). More interventional studies, using a cognitive rehabilitation approach, should be designed in selected phenotypes of MS patients, considering the different impacts of CI in the various MS phenotypes.

## 5. Conclusions

Home-based ICRP can improve cognitive functions in MS patients in the short term. A similar approach should be considered as a purposeful and regular program along with the main disease-modifying treatment for people with MS. This approach might be effective in preserving neuronal flexibility as well as in limiting the progression of cognitive dysfunction in MS.

## Figures and Tables

**Figure 1 jcm-11-03560-f001:**
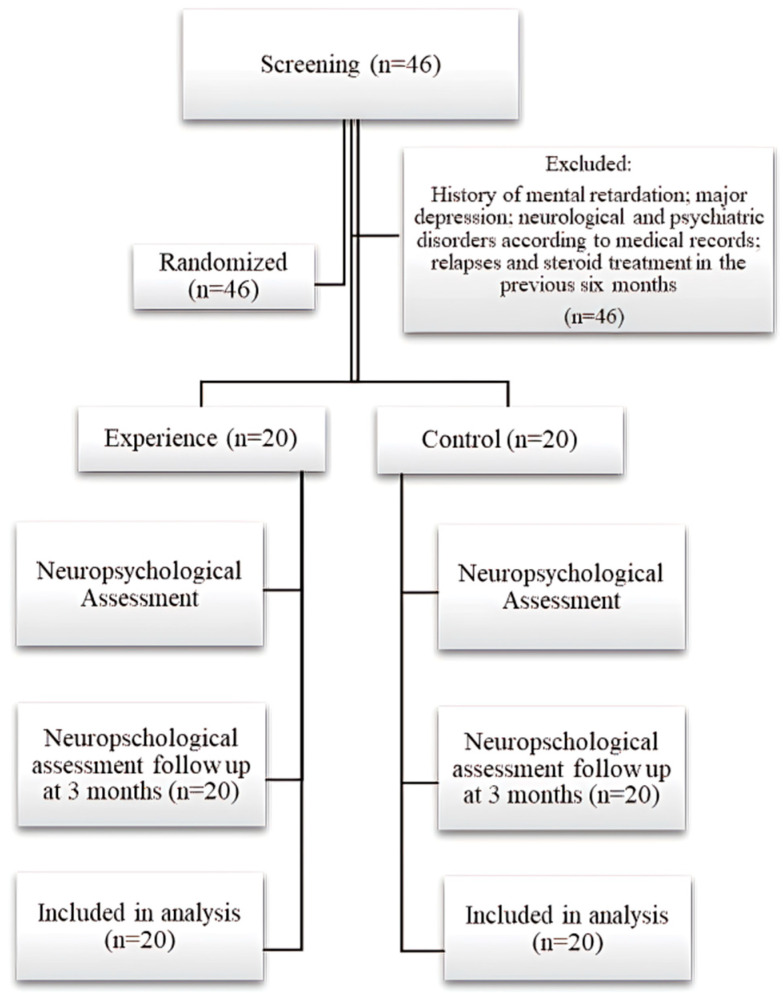
Flowchart of intervention.

**Figure 2 jcm-11-03560-f002:**
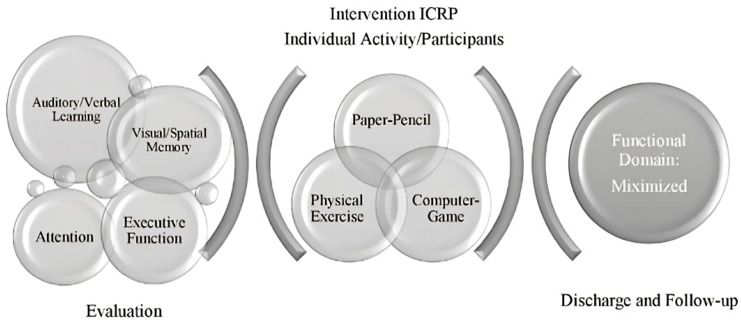
The Integrative Cognitive Rehabilitation Program Theoretical Model.

**Figure 3 jcm-11-03560-f003:**
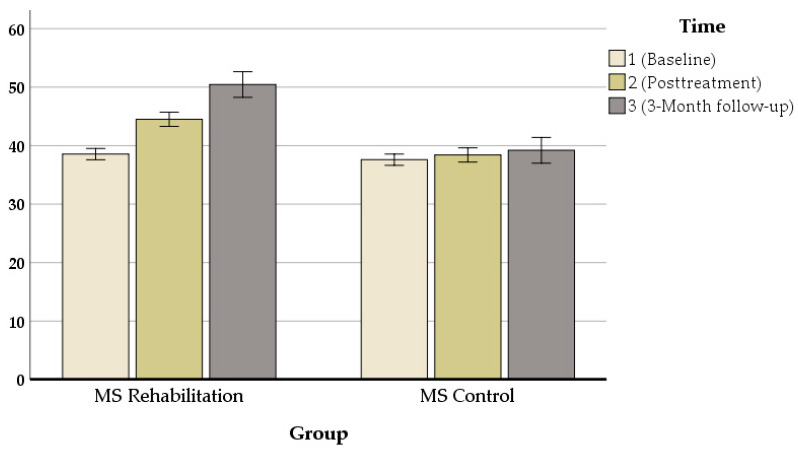
Composite auditory/verbal learning (CVLT)performance (z-scores) in the ICRP intervention group and control group at baseline and posttreatment.

**Figure 4 jcm-11-03560-f004:**
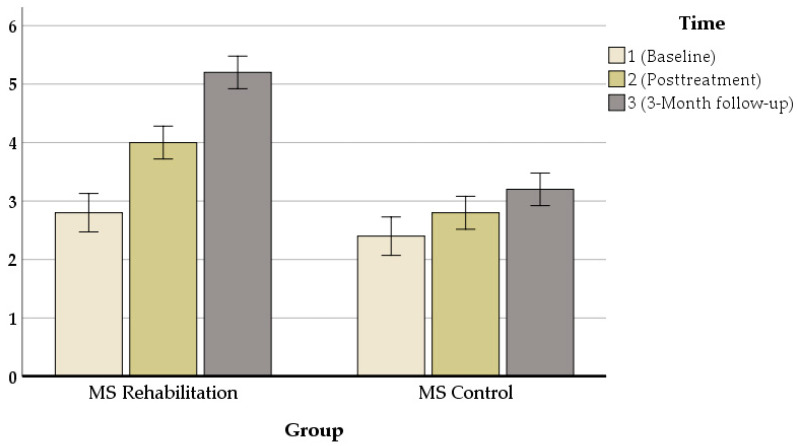
Composite visual/spatial memory (BVMTR) performance (z-scores) in the ICRP intervention group and control group at baseline and posttreatment.

**Figure 5 jcm-11-03560-f005:**
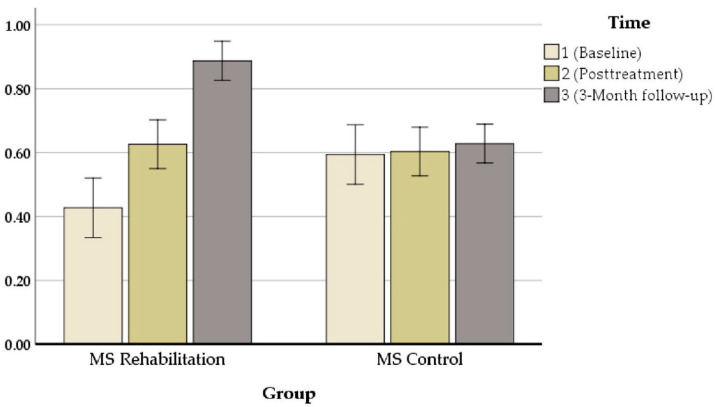
Composite attention performance (z-scores) in the ICRP intervention group and control group at baseline and posttreatment.

**Figure 6 jcm-11-03560-f006:**
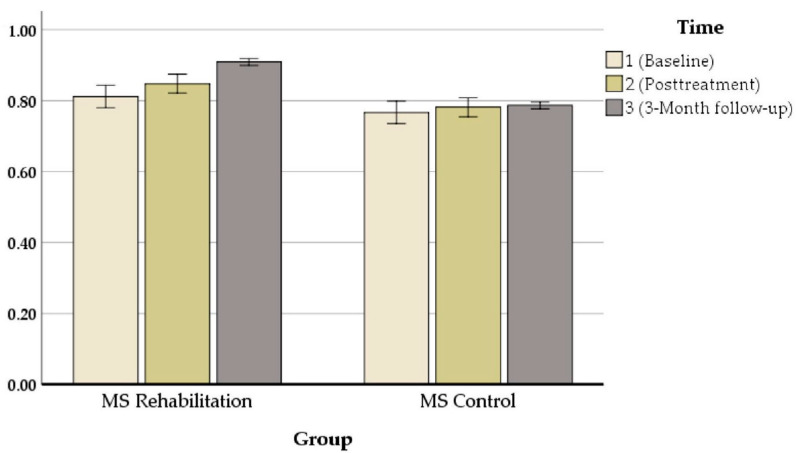
Composite executive function performance (z-scores) in the ICRP intervention group and control group at baseline and posttreatment.

**Table 1 jcm-11-03560-t001:** Demographic and clinical characteristics of the experimental and control groups.

Characteristic		Experimental Group (n = 20)	Control Group (n = 20)	*U/x* ^2^	*df*	*p*
**Sex**	Mean (SD) (95% CI)	1.3 ± 0.47	1.2 ± 0.41	149.00	39	0.988
	Female (%)	14 (70)	16 (80)			
	Male (%)	6 (30)	4 (20)			
**Age _years_**	Mean (SD) (95% CI)	34.15 ± 8.36	31.85 ± 8.25	0.512	1	0.774
	18–30 (%)	7 (35)	10 (50)			
	31–43 (%)	9 (45)	7 (35)			
	44–56 (%)	4 (20)	3 (15)			
**Disease duration _years_**	Mean (SD) (95% CI)	7.6 ± 4.32	7.8 ± 2.28	4.489	1	0.106
	2–8 (%)	15 (75)	14 (70)			
	9–14 (%)	3 (15)	3 (15)			
	15–21 (%)	2 (10)	3 (15)			
**History of MS disease in the family**	Mean (SD) (95% CI)	1.3 ± 0.47	1.2 ± 0.41	208.00	39	0.072
	With history (%)	6 (30)	4 (20)			
	Without history (%)	14 (70)	16 (80)			
**Phenotypes of MS**	Mean (SD) (95% CI)	1.75 ± 0.85	1.5 ± 0.82	4.305	1	0.116
	Relapsing-remitting (RRMS) (%)	10 (50)	14 (70)			
	Primary-progressive MS (PPMS) (%)	5 (25)	2 (10)			
	Secondary-progressive (SPMS) (%)	5 (25)	4 (20)			
**Disease-modifying therapies (DMTs)**	Interferon β-1b (%)	4 (20)	5 (25)	190.00	39	0.892
	Glatiramer acetate (%)	6 (30)	5 (25)			
	Ocrelizumab (%)	10 (50)	10 (50)			

Notes: SD; standard deviation; CI: confidence interval; df: degrees of freedom; N = total number of samples; U: Mann–Whitney U test; *x*^2^; Kruskal–Wallis.

**Table 2 jcm-11-03560-t002:** Performance on neuropsychological measures for the ICRP group and control group at baseline, posttreatment, and at 3-month follow-up.

Measure	Time Period	Experimental Group		Control Group	
		Mean	SD	Mean	SD
**CVLT**	T0	3/37	0/413	3/10	0/391
	T1	3/68	0/418	3/33	0/395
	T2	6/75	0/827	3/75	0/401
**BVMTR**	T0	2/80	0/89	2/40	0/50
	T1	4	0/77	2/80	0/41
	T2	5/20	0/76	3/20	0/41
**Attention**	T0	0/42	0/22	0/49	0/27
	T1	0/62	0/15	0/50	0/27
	T2	0/88	0/08	0/52	0/28
**Executive Function**	T0	0/81	0/09	0/76	0/03
	T1	0/84	0/07	0/78	0/03
	T2	0/90	0/007	0/78	0/02

Notes: All values are raw scores. T: Time, T0: baseline assessment; T1: posttreatment assessment; T2: 3-month follow-up assessment. CVLT: California Verbal Learning Test, BVMTR: Brief Visuospatial Memory Test.

**Table 3 jcm-11-03560-t003:** Two-way mixed-effect ANOVA for cognitive domain performance: time (within subjects’ factor) and patient group: (between subjects’ factor).

		Value	*f*	*p*-Value	Effect Size	Observed Power
**CVLT**	Time	0.363	66.681	0.000	0.637	1
	Time × group	0.494	38.816	0.000	0.505	1
**BVMTR**	Time	0.089	389.120	0.000	0.911	1
	Time × group	0.281	97.280	0.000	0.719	1
**Attention**	Time	0.276	48.623	0.000	0.724	1
	Time × group	0.339	36.124	0.000	0.661	1
**Executive Function**	Time	0.328	37.913	0.000	0.672	1
	Time × group	0.697	8.005	0.001	0.302	0.95

Notes: F: ANOVA Wilks’ Lambda (interaction effect), Effect size; (r) 0.1 small size; 0.3 medium size; 0.5 large size. Observed power (or post hoc power) is the statistical power of the test performed. CVLT: California Verbal Learning Test, BVMTR: Brief Visuospatial Memory Test.

**Table 4 jcm-11-03560-t004:** Comparison of neuropsychological test scores for the standard care MS control group at baseline and posttreatment.

	Baseline		Posttreatment		3-Month Follow-Up		Baseline versus Posttreatment *p*-Values	Effect Size (r)	Baseline versus Follow-Up *p*-Values
	Mean	Median	Mean	Median	Mean	Median			
**CVLT**	38.55	38	44.50	43.50	50.45	48.50	0.000 ***	0.996	0.000
**BVMTR**	2.80	2.50	4	3.75	5.20	5	0.000 ***	0.996	0.000
**Attention**	0.427	0.445	0.626	0.670	0.887	0.90	0.000 ***	0.944	0.000
**Executive Function**	0.811	0.840	0.847	0.875	0.909	0.910	0.000 ***	0.997	0.000

Notes: All values are raw scores (*** *p* < 0.001). Friedman’s nonparametric test was used for the comparison of medians between baseline, posttreatment, and follow-up. Wilcoxon test with Holm–Bonferroni correction used for pairwise comparisons. CVLT: California Verbal Learning Test, BVMTR: Brief Visuospatial Memory Test.

## Data Availability

The data presented in this study are available in this article.

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
