# Peer review of "Integrated Cognitive Rehabilitation Home-Based Protocol to Improve Cognitive Functions in Multiple Sclerosis Patients: A Randomized Controlled Study"

_jcm, 2022, doi:10.3390/jcm11123560_

Round 1
Reviewer 1 Report
This is a randomized, interventional study comparing the results on various cognitive endpoints between standard care (control group, n=20) and a cognitive rehabilitation program (interventional group, n=20). The cognitive rehabilitiation group showed improvements on various cognitive domains, so these results support the use of cognitive neurorehabilitation programs in patients with MS.
The methods are well explained and the results, plausible. There are minor concerns that do not prevent my recommendation for publication:
- Methods section: sample's size calculation is very briefly mentioned. Although the sample's size in this study is similar to others, it would be helpful to know how the authors have reached that n, based on what references, on what cognitive variable the calculation is made, if the calculation is based on mean differences (in which case, it would be interesting to know the variance of the quantitative variables that the control group is expected to have and the minimum value of the difference that the authors want to detect) or differences in proportions, and so.
- Methods section: MS clinical phenotypes are defined as relapsing-remitting, primary progressive and progressive-relapsing. These latter, progressive-relapsing, is no longer considered an MS phenotype as Lublin 2014. Progressive-relapsing was a subtype of progressive MS in which a primary progressive course was at first detected, but a relapse occurs thereafter, and was considered the less frequent of the MS phenotypes. I think the authors are referring to the secondary progressive MS; if not, they should explain why they would rather use an outdated MS classification.
- Table 1: the percentages of patients with or without history of familial MS seem to be reversed. I think it should state: with history 6(30) and without 14 (70).
Author Response
Reviewer 1
Comments and Suggestions for Authors
This is a randomized, interventional study comparing the results on various cognitive endpoints between standard care (control group, n=20) and a cognitive rehabilitation program (interventional group, n=20). The cognitive rehabilitation group showed improvements in various cognitive domains, so these results support the use of cognitive neurorehabilitation programs in patients with MS.
The methods are well explained and the results, plausible. There are minor concerns that do not prevent my recommendation for publication:
Authors: We thank the Reviewer#1 for her/his respectful comments.
- Methods section: sample size calculation is very briefly mentioned. Although the sample's size in this study is similar to others, it would be helpful to know how the authors have reached that n, based on what references, on what cognitive variable the calculation is made, if the calculation is based on mean differences (in which case, it would be interesting to know the variance of the quantitative variables that the control group is expected to have and the minimum value of the difference that the authors want to detect) or differences in proportions, and so.
Authors: We thank the Reviewer#1 for having raised this crucial point. We selected the sample size according to previous investigations using neuropsychological and MRI monitoring (Bonavita et al., 2015 doi: 10.1007/s00415-014-7528-z., Rilo et al., 2018 doi:10.1080/09638288.2016.1250168., Filippi et al., 2012 doi: 10.1148/radiol.11111299), also considering that a continuous cognitive rehabilitation intervention with this sample size (n=40) would be more feasible. The calculation of the sample size was based on the consideration that we expected to observe a minimum difference of cognitive functions between the two compared groups, such as in the study by Bonavita et al. (2015) and Rilo et al. (2018). We added this clarification in the revised text (page 3, line 107-108).
- Methods section: MS clinical phenotypes are defined as relapsing-remitting, primary progressive, and progressive-relapsing. The latter, progressive-relapsing, is no longer considered an MS phenotype as Lublin 2014. Progressive-relapsing was a subtype of progressive MS in which a primary progressive course was at first detected, but relapse occurs thereafter and was considered the less frequent of the MS phenotypes. I think the authors are referring to the secondary progressive MS; if not, they should explain why they would rather use an outdated MS classification.
Authors: Thanks for these valuable comments. We apologize for this inaccuracy. We specified in the text the included phenotypes of MS, represented by relapsing-remitting, primary progressive, and secondary progressive MS (page 3, line 113; reference Lublin et al. doi:10.1212/WNL.0000000000000560).
- Table 1: The percentages of patients with or without a history of familial MS seem to be reversed. I think it should state with history 6(30) and without 14 (70).
Authors: We apologize for this inaccuracy. The numbers of patients with familial MS were edited (Table 1).
Reviewer 2 Report
Dear Authors,
The manuscript presents improvement cognitive functions in multiple sclerosis patients after application the home-based integrated cognitive hehabilitation protocol. It is well written, however, the following issues need clarification.
1. Title
The title should clearly indicate that the Cognitive Rehabilitation protocol was home-based.
2. Introduction
The authors state in the introduction that ,,Patterns of cognitive dysfunction are heterogeneous in MS, and this can be noticed in the early stages of the disease as well.” However, they did not mention about differences in cognitive impairment between various MS variants. In the course of MS different processes are observed, including inflammation and neurodegeneration of varying severity. It has been proven that severe cognitive deficits are more frequently reported by patients with progressive disease types that those with RRMS, even after more than 10 years of disease (doi: 10.1111/ene.12715, doi: 10.1177/1352458516674367 ). Moreover, Dackovic et al. showed that among 168 MS patients cognitive deficit is most severe and most frequent in those with SPMS, followed by PPMS subjects and then CIS and RRMS patients (doi: 10.1007/s10072-016-2610-1). The authors should address these issues in the introduction. Furthermore, authors should point to the particular necessity of conducting such a study in a selected population of MS patients.
2. Methods
The refference for EDSS is lacking (lines 104-105). Why did the authors choose a cut-off point of at least 2 years of disease duration as the inclusion criterion? It has been shown that CI is present in all MS subtypes since the clinical onset, including up to 34.5% of CIS patients (doi: 10.1177/1352458516674367). Furthermore, which criteria of the clinical forms division proposed by Lublin and Reingold were taken into account by the authors because it is not clear to me and there is no reference (lines 103-104). There are no patients with secondary progressive MS among the study participants. Why were these patients excluded from the study? Moreover, there are no data on the number of pwMS taking disease-modifying therapies (DMTs) in both groups. DMTs have been shown to be effective in improving cognitive test performance in RRMS patients (doi: 10.1212/WNL.0000000000009522). Therefore, there should be comprehensive information on such therapies.
3. Dicussion
The authors note in the discussion that ,,It is worth mentioning that patients’ medical status was stable in both groups during the observation period”. As I understand, none of the participants experienced MS relapse. It is known that relapse may decline abruptly cognitive functions, therefore its occurrence during the examination should be a criterion for exclusion and not only additional information in the discussion.
4. Limitations
The main limitation of the study is the low number of respondents, which significantly reduces the statistical power of the results. Another limitation of the study is the lack of analysis of the influence of the disease variant and DMTs on the results. Noteworthy, patients with progressive MS type need more specialized CI management than RRMS subjects. Another issue is that the patients were monitor patients during the study was limited to "check-in" calls only.
Author Response
Reviewer 2
Comments and Suggestions for Authors
Dear Authors,
The manuscript presents improvement in cognitive functions in multiple sclerosis patients after the application of the home-based integrated cognitive rehabilitation protocol. It is well written; however, the following issues need clarification.
Authors: We appreciate your worth and useful comments.
- Title
The title should clearly indicate that the Cognitive Rehabilitation protocol was home-based.
Authors: We edited the title according to the comment of Reviewer#2 (Integrated Cognitive Rehabilitation home-based protocol to Improve Cognitive Functions in Multiple Sclerosis Patients: A Randomized Controlled Study).
- Introduction
The authors state in the introduction that, Patterns of cognitive dysfunction are heterogeneous in MS, and this can be noticed in the early stages of the disease as well.” However, they did not mention differences in cognitive impairment between various MS variants. In the course of MS different processes are observed, including inflammation and neurodegeneration of varying severity. It has been proven that severe cognitive deficits are more frequently reported by patients with progressive disease types that those with RRMS, even after more than 10 years of disease (doi: 10.1111/ene.12715, doi: 10.1177/1352458516674367 ). Moreover, Dackovic et al. showed that among 168 MS patients’ cognitive deficit is most severe and most frequent in those with SPMS, followed by PPMS subjects and then CIS and RRMS patients (doi: 10.1007/s10072-016-2610-1). The authors should address these issues in the introduction. Furthermore, the authors should point to the particular necessity of conducting such a study in a selected population of MS patients.
Authors: We thank the Reviwer#2 for this valuable comment. We clarified in the introduction that there are differences in cognitive impairment between various MS phenotypes, also citing the three mentioned papers (Planche et al., 2016; Ruano et al., 2017; Dackovic et al., 2016) (page 1 lines 41-42).
Moreover, we added in the limitation section a comment regarding the need for more interventional studies, using a cognitive rehabilitation approach, designed in selected MS phenotypes (page 13, lines 379-380).
- Methods
The reference for EDSS is lacking (lines 104-105).
Authors: According to this appropriate comment, we inserted a reference for the EDSS scale Kurtzke JF, 1983 Inojosa et al, 2020 (page 13, lines 115).
Why did the authors choose a cut-off point of at least 2 years of disease duration as the inclusion criterion? It has been shown that CI is present in all MS subtypes since the clinical onset, including up to 34.5% of CIS patients (doi: 10.1177/1352458516674367).
Authors: We thank the Reviewer#2 for this appropriate comment. We enrolled MS patients who were member in the MS Association of Alborz, Zahedan, and Zabol in Iran from at least two years and this choice was due to the need of including only definite diagnoses in heterogeneous groups of patients who came from different regions of Iran and were diagnosed and followed by various MS centers. We specified this point in the revised text (page 3, lines 103-106).
Furthermore, which criteria of the clinical forms division proposed by Lublin and Reingold were taken into account by the authors because it is not clear to me and there is no reference (lines 103-104).
Authors: We highlighted that RRMS, PPMS, and SPMS phenotypes were included (Lublin FD, Reingold SC, Cohen JA, et al. Defining the clinical course of multiple sclerosis: the 2013 revisions. Neurology. 2014;83(3):278-286. doi:10.1212/WNL.0000000000000560).
There are no patients with secondary progressive MS among the study participants. Why were these patients excluded from the study?
Authors: After having revised the manuscript according to the comments of Reviewer#1, we have better clarified the MS phenotypes included, which comprised also SPMS (please see the response to the second comment of the Reviewer#1).
Moreover, there are no data on the number of MS taking disease-modifying therapies (DMTs) in both groups. DMTs have been shown to be effective in improving cognitive test performance in RRMS patients (doi: 10.1212/WNL.0000000000009522). Therefore, there should be comprehensive information on such therapies.
Authors: We agree with Reviewer#2 that the therapeutic issue is another aspect to be monitored in both groups. In the revised manuscript we clarified that we included only patients undergone DMTs to have more homogeneity in therapeutic management (DMTs of the included MS patients were interferon beta-1b and Glatiramer acetate). However, this could be a bias considering that DMTs have been shown to improve cognitive performances in MS. We added these aspects to the methods (page 3 line 106) and limits (page 12, lines 375-377) sections. Also, DMTs were stable throughout the entire trial and, then, positive effects of DMTs on cognitive function could not be ruled out.
- Discussion
The authors note in the discussion that, it is worth mentioning that patients’ medical status was stable in both groups during the observation period”. As I understand, none of the participants experienced MS relapse. It is known that relapse may decline abruptly cognitive functions, therefore its occurrence during the examination should be a criterion for exclusion and not only additional information in the discussion.
Authors: We agree with Reviewer#2 regarding the inappropriate, brief note on the lack of observation of MS relapses during the follow-up period. We removed this statement from the revised version of the manuscript.
- Limitations
The main limitation of the study is the low number of respondents, which significantly reduces the statistical power of the results. Another limitation of the study is the lack of analysis of the influence of the disease variant and DMTs on the results. Noteworthy, patients with progressive MS type need more specialized CI management than RRMS subjects. Another issue is that the patients were monitor patients during the study were limited to “check-in" calls only.
Authors: Also, according to the previous comments of Reviewer#2, we added more limitations to the limit sections (page 12, lines 374-379). Among limits, we added also that the patients were mainly monitored mainly through “check-in" calls, although a cognitive therapist was available to patients every day of the intervention’s weeks to receive their functions reports. We clarified these points in the revised text (page 5 lines 190, 191; page 12, lines 379-380).
Round 2
Reviewer 2 Report
The authors responded to all of the indicated issues. However, the exact number of patients using DMTs has not yet been reported in the manuscript. The authors only mentioned in the response that ,,In the revised manuscript we clarified that we included only patients undergone DMTs to have more homogeneity in therapeutic management (DMTs of the included MS patients were interferon beta-1b and Glatiramer acetate)." Please enter these data in the table presenting the study group. So far, there is data on the family history of MS and there is no relevant information on DMTs. I also do not understand the phrase that ,,we included only patients undergone DMTs". Does this mean that patients with PPMS have also been treated with these drugs (INF or GA)? Please clarify this.
Author Response
We thank the Reviewer for remarking our mistake and we apologize for the oversight. As requested, we added data on DMTs in the table (page 7) and a description of DMTs (lines 228-229).
Progressive MS patients (PP and SP were treated with Ocrelizumab)